Comparative analysis of fecal DNA viromes in Large-billed crows and Northern ravens reveals diverse viral profiles

Dong Yonggang 1 2
Fan Sitong 2
He Shunfu 3
Zhao Wenxin 3
Lancuo Zhuoma 1
Sharshov Kirill 4 5
Li Ying yingli@126.com 1
Wang Wen 007cell@163.com 1
1 State Key Laboratory of Plateau Ecology and Agriculture, Qinghai University , Xining , China
2 College of Eco-Environmental Engineering, Qinghai University , Xining , China
3 Xining Wildlife Park of Qinghai Province , Xining , China
4 Federal Research Center of Fundamental and Translational Medicine , Novosibirsk , Russia
5 Novosibirsk State University , Novosibirsk , Russia
Breitbart Mya
Electronic publication date: 2025 Oct 15
Publication date: 2025
Volume: 13
Electronic Location ID: e20170
Received 2025 Jun 3; Accepted 2025 Sep 11
Copyright: ©2025 Dong et al.
Copyright year: 2025
Copyright holder: Dong et al.
License: This is an open access article distributed under the terms of the Creative Commons Attribution License, which permits unrestricted use, distribution, reproduction and adaptation in any medium and for any purpose provided that it is properly attributed. For attribution, the original author(s), title, publication source (PeerJ) and either DOI or URL of the article must be cited.
License URL: https://creativecommons.org/licenses/by/4.0/

Keywords: Fecal DNA viruses, Facultative scavenger birds, Northern raven, Large-billed crow, Viral metagenomics

Funding: The Program of Science and Technology International Cooperation Project of Qinghai Province 2022-HZ-812 The National Natural Science Foundation of China and Russian Foundation for Basic Research Cooperative Exchange Project 32111530018 This research was funded by the Program of Science and Technology International Cooperation Project of Qinghai Province (grant No. 2022-HZ-812), the National Natural Science Foundation of China and Russian Foundation for Basic Research Cooperative Exchange Project (grant No. 32111530018). The funders had no role in study design, data collection and analysis, decision to publish, or preparation of the manuscript.

==============================
As facultative scavenger birds, crows carry various parasites, viruses, and bacteria, making them significant infection hosts and transmission vectors. In this study, we employed viral metagenomics to enrich viral particles from three fecal samples of the Northern ravens (Corvus corax) and four fecal samples of the Large-billed crows (Corvus macrorhynchos). Viral DNA was then extracted, and seven sequencing libraries were constructed. The composition and characteristics of the DNA viromes in the feces of these two facultative scavenging bird species were analyzed using the Illumina NovaSeq platform (PE150 mode). The results showed that the fecal DNA viruses carried by Northern ravens mainly belonged to Parvoviridae (31.49%), Caudoviricetes_Unclassified (21.91%), Microviridae (21.57%), and Genomoviridae (18.2%), while those carried by Large-billed crows were predominantly Genomoviridae (29.7%), Parvoviridae (26.15%), and Caudoviricetes_Unclassified (22.15%). Diversity analysis using Richness, Shannon, and Simpson indices showed no significant differences in viral composition between the two crow species. Additionally, principal coordinate analysis (PCoA) (F = 1.079, P = 0.155) and non-metric multidimensional scaling (NMDS) (F = 1.079, P = 0.154) analyses demonstrated no distinct separation between the two groups. Moreover, the KEGG-enriched pathways in both crow species were primarily associated with metabolic and genetic information processing functions. The selection of the Large-billed crows and Northern ravens in this study was based on their widespread distribution, close association with human settlements, and distinctive scavenging behavior. Comparative analysis of the diversity and composition of their DNA viral communities offers a basis for evaluating the zoonotic risks associated with these scavenger birds.

Introduction

Viruses, as one of the simplest and most abundant biological entities on Earth, require parasitic existence within living cells for survival, possessing merely a single type of nucleic acid (either DNA or RNA) (Pellett, Mitra & Holland, 2014). These organisms are ubiquitously found across diverse environments, encompassing animals, plants, microorganisms, humans, aquatic systems, and soils (Labadie et al., 2020). Zoonotic diseases are defined as those that can naturally be transmitted between vertebrates and humans (Rupasinghe, Chomel & Martinez-Lopez, 2022). Most emerging infectious diseases (EIDs), accounting for about 60.3%, fall under the category of zoonoses, with approximately 71.8% of these cases stemming from wildlife (Jones et al., 2008). The pathogens responsible for these diseases, including bacteria, viruses, parasites, and fungi, can be transmitted from animals to humans either directly or indirectly through food, water, and environmental pathways, potentially resulting in infections that range from mild to severe, and even fatal (McArthur, 2019; Kinnunen et al., 2022). Owing to their remarkable ability to mutate, viruses are particularly prone to crossing species boundaries to infect humans, thereby serving as principal pathogens in zoonotic infectious diseases.

Research indicates that viruses harbored by wild birds are intimately associated with the spread and emergence of numerous human and animal infectious diseases (Berg et al., 2001; Poulson et al., 2024). During pathogen detection of Avian Influenza Virus (AIV) in wild birds near an Italian farm, researchers identified two H7N1-positive samples from 103 tested (Capua et al., 2000). In April to May 2005, a highly pathogenic H5N1 avian influenza outbreak was reported at Qinghai Lake, China. Among the approximately 1,500 wild birds found dead, 90% were bar-headed geese. The H5N1 subtype avian influenza virus was successfully isolated and characterized from these infected birds, including the bar-headed geese (Chen et al., 2005). Additional notable pathogens like Avian Poxvirus (Jarvi et al., 2008), West Nile Virus (LaDeau, Kilpatrick & Marra, 2007; Bakonyi et al., 2013), St. Louis Encephalitis Virus (Reisen et al., 2001), and Japanese Encephalitis Virus (Jamgaonkar et al., 2003) have also been associated with wild birds. Consequently, monitoring and comprehending the viruses carried by wild birds hold substantial importance for the prevention and control of viral epidemics.

Dietary preferences in birds may correlate closely with the types of viruses they harbor, suggesting that carnivorous birds may carry viruses associated with their predatory activities (Vidaña et al., 2020), whereas herbivorous or omnivorous species might host viruses linked to their dietary resources (Wang et al., 2019). Scavenging birds act as “sanitizers” within ecosystems by consuming large quantities of carcasses and organic debris, thus diminishing pathogen accumulation and mitigating the risk of interspecies disease transmission (OBryan et al., 2018). Despite their critical ecological functions, these birds, including obligate scavengers like vultures and facultative scavengers like crows, remain underappreciated and sometimes perceived as threats (Lambertucci et al., 2021). Recently, populations of obligate scavenging vultures have markedly decreased, largely due to toxins (particularly diclofenac and poisons) present in their diet (Nambirajan et al., 2018; Cuthbert et al., 2014). Further contributing factors include hunting, habitat loss, reduced food availability, and random events (Garcia-Jimenez et al., 2022; Buechley & Sekercioglu, 2016). The decline in obligate scavengers would inevitably offer additional food resources to other scavengers, such as crows, which as facultative scavengers, may come into contact with a greater array of bacteria, fungi, and viruses when confronted with more carcasses. Given that their digestive efficiency is not as robust as that of obligate scavengers, there is an elevated risk of pathogen infection and disease transmission. Although scavenging birds generally possess adaptive physiological mechanisms (like specialized facial features and gut microbiota) to counteract pathogens in their diet, thereby minimizing their own infection risk (Zepeda Mendoza et al., 2018; Hu et al., 2024a), they can still serve as hosts for specific pathogens, particularly those capable of surviving in harsh conditions (Zhai et al., 2023). Furthermore, crows, ubiquitous in urban settings and renowned for their intelligence and social behavior, tend to congregate (Ericson et al., 2005). Studies have shown that corvids, exemplified by the rook (Corvus frugilegus), which inhabit landfill sites, exhibit a high carriage rate of E. coli and a range of other pathogens (Malekian, Shagholian & Hosseinpour, 2021). Serving as vectors for zoonotic pathogen transmission, they are exposed to a wide spectrum of zoonotic pathogens (Nabi et al., 2021).

It is noteworthy that viruses harbored by wild birds may pose potential threats to human and animal health, yet the microbial communities within their intestines and their interactions should not be ignored either. Particularly, bacteriophages are crucial in shaping the composition and evolution of bacterial communities in natural environments (Chevallereau et al., 2022). Research on bacteriophages in mice indicates that they can not only directly suppress their targeted bacteria but also induce cascade effects on non-targeted bacteria via inter-bacterial interaction networks, thus systematically remodeling the structure and metabolic profile of the gut microbiota (Hsu et al., 2019). Additionally, bacteriophages can disseminate antibiotic resistance genes (ARGs) through mechanisms of horizontal gene transfer (Hu et al., 2024b). The dissemination of antibiotic resistance genes (ARGs) represents a significant public health crisis, as ARGs spread swiftly across the gut microbiome via horizontal gene transfer (HGT), causing many pathogens to develop multi-drug resistance, thereby directly compromising the effectiveness of current antibiotics. Globally, around 5 million deaths in 2019 were linked to antimicrobial resistance (AMR) (Zhang et al., 2022). Given that crows frequently consume food remnants or animal carcasses rich in bacteria, including antibiotic-resistant strains, from areas influenced by human activities (like garbage dumps and farm peripheries), their gut environment could potentially serve as a hotspot for interactions among bacteria, bacteriophages, and the ARGs they harbor. Understanding the composition of DNA viruses, particularly bacteriophages, and the potential ARGs they carry within the crow gut microbiota is essential for evaluating their roles in the dissemination of antibiotic resistance genes in the environment and the associated public health risks.

Viral metagenomics, also known as metaviromics, constitutes a specialized domain within metagenomics, focusing on the study of the genetic material of all viruses present in a sample. This approach enables the analysis of all viral nucleic acids in a given environmental sample, facilitating rapid assessment of the viral community and the discovery of potentially novel viruses (Nogueira et al., 2022). Compared to conventional virological methods, this technique offers advantages in terms of speed, accuracy, and scalability, enabling the detection of diverse pathogens from various sample sources. Recent research has highlighted the importance of evaluating the potential risks that viruses pose to organisms and environments in complex and varied samples, exploring virus-virus interactions, virus-environment interactions, and investigating viral taxonomy, relative abundance, and functional attributes (Sommers et al., 2021). Despite the use of this technology to profile viral communities across various species, a systematic understanding of the virome diversity, dynamic variations, and ecological roles of Corvidae species which are widely distributed and closely linked to human activities, is still insufficient. DNA viruses in avian species, particularly those associated with scavenger birds, have not yet been thoroughly investigated, despite potentially playing a significant role in the transmission of zoonotic diseases. Moreover, both the Large-billed crow and the Northern raven are omnivorous birds with a wide range of dietary preferences, including insects, small animals, fruits, seeds, and carrion (Wang et al., 2025).

Based on the above background, this study posits the following hypothesis: the two widespread and human activity-associated corvid species, the Northern raven and the Large-billed crow, might harbor a variety of DNA viruses (including but not limited to those linked with zoonotic diseases), and the viral communities within these two species could display notable differences. Owing to their omnivorous nature (particularly scavenging) and significant overlap with human habitats, crows are likely exposed to diverse ARGs, acting as important carriers for their dissemination. Considering the social behavior traits within crow populations, such as flocking, these characteristics might further enhance the transmission of viruses and ARGs between individuals and potentially across entire populations. Hence, this study utilizes viral metagenomics to investigate the intestinal DNA viral communities of two Corvidae species, the Northern raven and the Large-billed crow, with the goal of comprehensively characterizing the types and abundance of DNA viruses they harbor, performing comparative analyses, and providing scientific support for the prevention of zoonotic diseases.

Materials & Methods

Ethics approval

This study followed the Guidelines for the Care and Use of Experimental Animals issued by the Ministry of Science and Technology of the People’s Republic of China (approval number: 2006-398), and was reviewed and approved by the Ethics Committee of Qinghai University. All experimental procedures adhered to the 3R principles (replacement, reduction, refinement), employing non-invasive sampling methods to minimize potential impacts on wildlife. Fresh fecal samples from Large-billed crows and Northern ravens were opportunistically collected in the wild, without capturing, handling, or disturbing the birds. As this study involved only field-collected biological material and did not require direct interaction with live animals, it also followed taxon-specific ethical guidelines for avian observational and non-invasive sampling research. All local regulations and laws were adhered to.

Data availability statement

The raw sequence data reported in this paper have been deposited in the Genome Sequence Archive in National Genomics Data Center, China National Center for Bioinformation/Beijing Institute of Genomics, Chinese Academy of Sciences (GSA: CRA025033) that are publicly accessible at https://ngdc.cncb.ac.cn/gsa.

Sample collection

This research centers on two widespread corvid species on the Qinghai-Tibet Plateau: the Large-billed crow (abbreviated as DZ) and the Northern raven (abbreviated as DY), both of which exhibit extensive distribution and strong adaptability. These birds frequently inhabit high-altitude grasslands, shrublands, and areas close to human settlements, making them significant species in their ecosystems with relatively stable populations. The fecal samples for the DZ group (n = 4) were collected in Zeku County, Qinghai Province, situated in the northeast of the Qinghai-Tibet Plateau, a typical cold region, at coordinates 101°55′43″E, 35°15′21″N, and an elevation of 2,964 m. These samples were collected in August. For the DY group (n = 3), samples were collected in Yushu City, Qinghai Province, which lies in the central Qinghai-Tibet Plateau, featuring persistent snow cover and typical plateau cold conditions, at coordinates 96°59′35″E, 32°59′25″N, and 3,835 m above sea level. These samples were collected in May (Fig. 1). Crow species were identified based on morphological characteristics, including body size, beak type, and feather color, prior to sample collection.

Figure 1 Images of fecal samples from Large-billed crows and Northern ravens.

The figure shows the sampling locations and sample sizes for Large-billed crows and Northern ravens within the Qinghai Province of China.

Once the target birds were located in the wild, researchers observed and assessed their physical condition remotely, ensuring not to interfere with their natural behaviors. Assessment criteria encompassed normal flying ability (without flight impediments or unusual postures), normal feeding or alertness behaviors, responsiveness, clean plumage without noticeable defects or soiling, proportionate body form without signs of thinning, and absence of visible injuries or excretions (e.g., ocular or nasal discharges). Only individuals without any observable abnormal behavior or clinical symptoms were deemed apparently healthy. Upon confirming the apparent health of the target individual, continuous observation was carried out until natural defecation occurred, with researchers tracking the individual throughout until it left the defecation spot, verifying that the fecal sample originated from the targeted individual. Immediately afterwards, the site was accessed to collect fresh fecal samples using disposable sterile gloves and tweezers, taking care to avoid contact with the ground or other possible contamination sources. Each sample was promptly transferred to sterile microcentrifuge tubes (EP) and frozen in liquid nitrogen tanks. Additionally, gloves and sampling instruments were replaced immediately after each sample collection to strictly prevent cross-contamination. Throughout the collection process, speed was ensured to minimize exposure to air and environmental factors. All samples were then uniformly stored in −80 °C ultra-low temperature freezers upon returning to the lab, pending further viral particle enrichment and metagenomic analysis.

Concentration and enrichment of virus particles and extraction of DNA

The virus particles were concentrated and enriched using the methods outlined earlier (Sarker, 2021). To concentrate and enrich viral particles, fecal samples were mixed with sterile PBS in a 1:5 volume ratio and homogenized. Following three freeze-thaw cycles, the mixture underwent centrifugation at 12,000×g for 5 min at 4 °C to remove the pellet. The supernatant was subsequently filtered through a 0.45 µm followed by a 0.22 µm filter, and then added to a solution containing 28% (w/w) sucrose. The sample was then subjected to ultracentrifugation using an HIMAC CP ultracentrifuge (Hitachi, Tokyo, Japan) at 160,000×g for 2 h at 4 °C. Post-centrifugation, the supernatant was discarded, and the residual pellet was resuspended in 200 µL of sterile PBS. To degrade unprotected nucleic acids, a DNase mix consisting of Turbo DNase (Ambion, Austin, TX, USA), Baseline-ZERO (Epicenter), and Benzonase (Novagen, Darmstadt, Germany), together with RNase from Fermentas, was applied to the suspension and incubated at 37 °C for 2 h. The nuclease reaction was inactivated by adding two µL of 500 mM EDTA and heating at 75 °C for 10 min. Total nucleic acid extraction was conducted following the protocol provided by the manufacturer using the QIAamp Viral DNA Mini Kit (Qiagen, Hilden, Germany). Following this, the extracted nucleic acids underwent whole genome amplification using Qiagen’s REPLI-g Cell WGA & WTA Kit (catalog number 150,054). The quantity and quality of the purified amplification products were evaluated using a NanoDrop spectrophotometer (Thermo Fisher Scientific, Waltham, MA, USA) and 1.5% agarose gel electrophoresis. In total, seven sequencing libraries were prepared, four from Large-billed crows and three from Northern ravens, with each library comprising DNA samples from distinct individuals.

Library preparation and high-throughput sequencing

Metagenomic libraries were constructed using the Illumina TruSeq™ Nano DNA Sample Preparation Kit (San Diego, CA, USA) with one µg of total DNA. The process followed the NGS protocol for DNA end repair, A-base addition, and ligation of NGS indexed adapters. Target DNA fragments of approximately 400 bp were selected using a 2% low-range ultra-pure agarose gel, followed by amplification through 15 PCR cycles using Phusion DNA Polymerase (NEB). Sequencing was conducted by Guangdong Magigene Biotechnology Co., Ltd. (Guangzhou, China). All samples were sequenced on a next-generation sequencing platform in paired-end 150 bp (PE150) mode.

Bioinformatics analysis

The sequencing data was first subjected to quality control using Trimmomatic (Bolger, Lohse & Usadel, 2014), trimming and filtering the raw paired-end reads with parameters (SLIDINGWINDOW:4:15 MINLEN:75). Quality-controlled reads were subsequently assembled using MegaHit with the parameter (–min-contig-len 500) to generate contigs for each sample (Li et al., 2015). Following this, the assembled genomes were annotated. To ensure annotation accuracy, sequences exceeding 2,000 bp in length were chosen and identified as viral using VirFinder (Ren et al., 2017) (criteria: score > 0.7 and p-value < 0.05), VirSorter2 (Guo et al., 2021), and the IMG/VR database. Viral operational taxonomic units (vOTUs) were clustered using Mummer software, where candidate viral sequences were compared, requiring intra-cluster sequence similarity greater than 95% and coverage of more than 85% of the full sequence length. The longest vOTU in each cluster was considered the representative sequence (Roux et al., 2019). Virus species annotation and host prediction were carried out using geNomad and PhaBOX tools. Gene predictions were made using METAProdigal (Hyatt et al., 2012), followed by BLASTX comparisons against the NCBI non-redundant protein (NR), STRING, eggNOG (Huerta-Cepas et al., 2016), and KEGG databases to identify proteins with high sequence similarity to the query transcripts, achieving functional annotations with an E-value threshold usually set below 1.0 ×10−5. Unique assembled transcripts were annotated using the BLAST2GO program to describe biological processes, molecular functions, and cellular components. Metabolic pathways were analyzed using the Kyoto Encyclopedia of Genes and Genomes (KEGG). α-Diversity was analyzed using Mothur v.1.21.1 (Schloss et al., 2009), revealing indices such as richness and the Shannon diversity index. β-Diversity analysis was performed using the vegan package in R, conducting viral community similarity analyses via 999 permutations of the Bray–Curtis distance matrix. Correlations between viruses and other elements (such as other species and metabolites) were analyzed using Spearman’s rank correlation coefficient (Spearman’s r > 0.6) with statistical significance (P < 0.05) using relevant R packages. Gephi was utilized to create correlation heatmaps and network diagrams for visualizing these relationships. PCA, PCoA, and NMDS analyses were also conducted using the vegan package (2.6-4) in R (Oksanen et al., 2012). Differences between groups were evaluated using the Kruskal–Wallis non-parametric test to determine overall group differences, with significance indicated by * (P < 0.05). Furthermore, to perform phylogenetic analysis and construct the evolutionary tree of the viral sequences, we first employed the ViPTree tool (v4.0; http://www.genome.jp/viptree) to identify appropriate reference sequences for the detected viral genomes (Nishimura et al., 2017), which were subsequently retrieved from NCBI. Sequence alignment was carried out using MAFFT (v7.525) under default settings. Subsequently, phylogenetic inference was performed with IQ-TREE (v2.3.4), in which the optimal evolutionary model was selected automatically via ModelFinder Plus (MFP). Node support was evaluated using 1000 ultrafast bootstrap replicates.

Results

Overview of virome data

In this study, sequencing of four virome samples from DZ and three from DY using the Illumina NovaSeq PE150 platform resulted in a total of 502,883,606 raw reads (Table S1). Following quality filtering and depletion of host-derived sequences, a total of 218,918,824 reads were obtained (Tables S2, S3). The remaining reads were assembled into contigs longer than 500 bp, yielding a total of 349,657 contigs (Table S4). Considering the limited information contained in shorter sequences, those ≥2,000 bp were chosen for virus identification using tools including VirSorter2, VirFinder, deepVirFinder, and IMG/VR, leading to the identification of 78,568 viral sequences (Fig. S1, Table S5) as the final set of viral candidates. Following integration, filtering, and deduplication of the candidate viral sequences, 28,563 unique vOTUs were identified, totaling 143,349,097 bases, with an average length of 5,018.69, GC content of 48.17%, ranging in length from 212 to 15,290, and an N50 length of 5,649. Assessment of vOTU completeness using CheckV revealed that 3.51% of the vOTUs were of high quality, whereas 61.63% were of low quality (Fig. S2).

Composition of the viral community

Taxonomic annotation of the assembled viral sequences using geNomad showed that the DNA viral communities in both the DZ and DY viromes were predominantly dominated by single-stranded DNA viruses. The DZ virome was annotated into seven phyla, 13 classes, and 46 families, with the kingdom-level composition primarily comprising Monodnaviria (77.43%), representing single-stranded DNA viruses, and Duplodnaviria (22.52%), representing double-stranded DNA viruses. At the phylum level, the top five viral phyla in the DZ virome were Cressdnaviricota (41.20%), Cossaviricota (26.25%), Uroviricota (22.53%), Phixviricota (8.59%), and Hofneiviricota (1.43%), while other phyla accounted for less than 1% (Fig. 2A). At the family level, the top ten viral families were Genomoviridae (29.7%), Parvoviridae (26.15%), Caudoviricetes_Unclassified (22.15%), Microviridae (8.3%), Cressdnaviricota_Unclassified (3.1%), Cremevirales_Unclassified (1.8%), Mulpavirales_Unclassified (1.7%), Smacoviridae (1.68%), Cirlivirales_Unclassified (1.43%), and Inoviridae (1.4%) (Fig. 2B). The DY virome was annotated into seven phyla, 14 classes, and 37 families, with the kingdom-level composition again dominated by Monodnaviria (77.81%) and Duplodnaviria (22.15%). At the phylum level, the top five viral phyla in the DY virome were Cossaviricota (32.68%), Phixviricota (24.79%), Uroviricota (22.15%), Cressdnaviricota (20.04%), and Hofneiviricota (0.3%), with all other phyla accounting for less than 1% (Fig. 2A). At the family level, the top ten viral families were Parvoviridae (31.49%), Caudoviricetes_Unclassified (21.91%), Microviridae (21.57%), Genomoviridae (18.2%), Petitvirales_Unclassified (3.22%), Piccovirales_Unclassified (0.87%), Geplafuvirales_Unclassified (0.61%), Cressdnaviricota_Unclassified (0.4%), Cirlivirales_Unclassified (0.36%), and Inoviridae (0.3%) (Fig. 2B).

Figure 2 Abundance composition and differential analysis of DNA viral communities in Northern ravens and Large-billed crows.

(A) Phylum-level viral abundance and composition, (B) family-level viral abundance and composition, (C) phylum-level intergroup differences, (D) family-level intergroup differences.

Analysis based on phylum-level taxonomic annotations indicated differences in the species composition of viral communities between the DZ and DY groups. Cressdnaviricota was dominant in DZ (41.20%), whereas Cossaviricota predominated in DY (32.68%). And the inter-group difference test at the phylum level showed no significant difference between the two crow species (Fig. 2C). Notably, unclassified viruses (e.g., Caudoviricetes_Unclassified and Petitvirales_Unclassified) constituted a substantial proportion. Microviridae exhibited higher abundance in DY, while Cressdnaviricota_Unclassified showed a slightly higher proportion in DZ. The inter-group difference test for the top five viral families at the family level revealed no statistically significant differences between the two crow species (Fig. 2D). Diversity analysis using Richness, Shannon, and Simpson indices also revealed no statistically significant differences in viral diversity between the two species (Fig. 3A). Additionally, non-metric multidimensional scaling (NMDS) and principal coordinate analysis (PCoA) results (Figs. 3B, 3C) demonstrated that the viral communities of the two crow species lacked evident clustering trends within groups and clear separation trends between groups.

Figure 3 Analysis of alpha and beta diversity of DNA viral communities in Northern ravens and Large-billed crows.

Comparison of Richness, Shannon, and Simpson indices between the two groups (A); NMDS analysis based on Bray-Curtis distances (B); and principal coordinates analysis (PCoA) (C).

Analysis of phage composition

Annotation of phage sequences revealed that most phages in both groups belonged to the phylum Uroviricota, along with several unclassified viral groups. In the DZ group, the five most abundant phage families were Mesyanzhinovviridae (28.36%), Salasmaviridae (27.34%), Casjensviridae (12.77%), an unclassified phage genome (NC_048198), which could not be assigned to any known viral family and represented 7.59%, and Drexlerviridae (5.47%). All other families had relative abundances below 5%. In the DY group, phages also predominantly came from the Uroviricota phylum (95.81%) and other unclassified viral phyla. The top five most abundant families were Guelinviridae (59.91%), Salasmaviridae (9.9%), Drexlerviridae (8.6%), Chaseviridae (5.17%), and Casjensviridae (3.88%), with all other families contributing less than 3% (Fig. 4A). A box plot analysis comparing the top five families between groups (Fig. 4B) showed significant enrichment of Guelinviridae in the DY group, indicating a significant difference. Concerning the lifestyle of phages, the PhaBOX tool identified 11,199 phage sequences in the DZ virome, with 37.5% being lytic phages and 18.3% being temperate phages; in the DY virome, it identified 5,137 phage sequences, with lytic phages accounting for 33.9% and temperate phages for 26.1%. Furthermore, the VIBRANT tool was employed to analyze phage lifestyles, identifying 13,324 phage sequences in the DZ virome, where lytic phages dominated at 96.2%, and in the DY virome, 5,379 sequences were identified, also dominated by lytic phages at 95.3%. The identification of nucleocytoplasmic large DNA viruses (NCLDVs) in both viral groups was performed using VirSorter2, and the results indicated that 668 NCLDV sequences were identified in the DZ virome and annotated using geNomad, showing that these NCLDV species primarily belonged to various families within the Uroviricota phylum, with 235 sequences remaining unannotated. In the DY virome, 419 NCLDV sequences were identified, with annotations similar to those in the DZ virome, and 183 sequences remained unannotated.

Figure 4 Analysis of the composition and differences of bacteriophages within the DNA viruses of Northern ravens and Large-billed crows.

The bar chart illustrates the diversity and abundance of bacteriophage types present in each group (A). The box plot highlights the intergroup variation among the top five most abundant bacteriophage families (B).

Functional annotation

Gene annotations of vOTUs fragments were conducted using the metaProdigal software, identifying a total of 224,144 coding sequences across the seven viromes from both DZ and DY. The total sequence length amounted to 131,655,054 bp, with an average of 1.56 genes per kilobase and an average sequence length of 587 bp. To fully leverage the complementary strengths of various databases, this study utilized multiple databases for functional annotation. Specifically, the protein sequences of predicted genes were aligned with NR, Swiss-Prot, eggNOG, KEGG, GO, and VFDB databases to obtain detailed annotation information for these predicted genes and to understand the functional characteristics of the corresponding viral genes.

This study focuses on the functional analysis of annotated vOTU genes using the COG, GO, and KEGG databases. Initially, a functional annotation analysis of samples from both DZ and DY groups was conducted using the COG database, identifying 22 COG functional categories. In the DZ group (Fig. 5A), the main functional categories included cell wall/membrane/envelope biogenesis (COG category M), replication, recombination and repair (COG category L), amino acid transport and metabolism (COG category E), and coenzyme transport and metabolism (COG category H). In the DY group, 22 primary functional categories were identified, with core functions related to replication, recombination and repair (COG category L), cell wall/membrane/envelope biogenesis (COG category M), translation, ribosomal structure and biogenesis (COG category J), and posttranslational modification, protein turnover, and chaperones (COG category O) (Fig. 5B). Subsequently, the KEGG database was utilized to annotate the signaling pathways involving vOTU genes, identifying 20 critical pathways (Fig. 5C), among which the most active was the DNA replication pathway, followed by metabolic pathways, homologous recombination, and mismatch repair pathways. The KEGG pathway Level 2 heatmap (Fig. 5D) reveals that the top five enriched pathways in the DZ group are replication and repair, cell growth and death, signal transduction, aging, and prokaryotic cell communities. In the DY group, the top five pathways are replication and repair, global and overview maps, nucleotide metabolism, coenzyme and vitamin metabolism, and cell growth and death. It was observed that functions related to metabolism and genetic information processing dominate in both the DZ and DY viromes. The foremost three metabolic categories include global and overview metabolism, nucleotide metabolism, and cofactor and vitamin metabolism, whereas genetic information processing is primarily concerned with replication and repair.

Figure 5 Functional annotation analysis of the virome.

Results of the Cluster of Orthologous Groups (COG) functional annotation for Large-billed crows (A) and Northern ravens (B). A bar chart showing the abundance of the top 20 significant signaling pathways in KEGG (C), functional pathways at KEGG Level 2 for both groups (D), and annotation results at the second level of Gene Ontology (GO) (E).

To investigate the biological processes, molecular functions, and cellular components involving vOTU genes, a functional enrichment analysis was performed using the GO database, as shown in Fig. 5E. In terms of biological processes, vOTU genes predominantly participate in metabolism, cell growth and development, and interspecies communication. From a molecular function standpoint, these genes are primarily linked to catalytic activity, molecular binding, and small molecule activity. Regarding cellular localization, vOTU genes are mainly located within viral and cellular environments. CAZy database annotation showed that six enzyme types were identified in the two virome gene sets, including a total of 2,715 carbohydrate-active enzymes (CAZymes): 15 auxiliary activity enzymes (AA), 574 carbohydrate-binding modules (CBM), 192 carbohydrate esterases (CE), 1,694 glycoside hydrolases (GH), 217 glycosyltransferases (GT), and 23 polysaccharide lyases (PL) (Fig. 6A). In the DZ group, AA and GT were more abundant compared to the DY group, while the DY group showed higher abundance in the other four enzyme types. Nevertheless, there were no significant differences between the two groups (Fig. 6B). Based on the SARG database (v2.3) annotation, the DZ group contained 13 types of antibiotics, with polymyxin (69.92%) and MLS (14.88%) being predominant, while other types accounted for less than 5% (Fig. 6C). The DY group had 12 types of antibiotics with relatively even distribution: trimethoprim (18.53%), MLS (17.11%), multidrug (14.4%) among others (Fig. 6D). Furthermore, 34 unique antibiotic resistance genes (ARGs) were identified in the DZ group, compared to 39 in the DY group (Table S6).

Figure 6 Comparative analysis of CAZyme distribution and antibiotic composition in Large-billed crows and Northern ravens.

A bar chart displays the number of CAZymes detected in Large-billed crows and Northern ravens (A), where AA refers to Auxiliary Activities; CBM refers to Carbohydrate-Binding Modules; CE refers to Carbohydrate Esterases; GH refers to Glycoside Hydrolases; GT refers to Glycosyl Transferases; and PL refers to Polysaccharide Lyases. Boxplots show the differences in CAZymes between the two groups (B). Pie charts illustrate the structural composition and abundance of antibiotics in the Large-billed crow group (C) and the Northern raven group (D).

Virus-host prediction and phylogenetic analysis

Predicting viral hosts provides insight into their functional roles and ecological impacts. The results revealed that in the DZ group, the major bacterial hosts of the top five phages with highest host counts were Chlamydia pneumoniae, Geobacillus kaustophilus, and Staphylococcus saprophyticus (Fig. 7A). In the DY group, additional host species were identified, including Staphylococcus saprophyticus and Lactobacillus fermentum (Fig. 7B). To more accurately reflect the genetic diversity of viruses, viral sequences from five families (Genomoviridae, Parvoviridae, Smacoviridae, Inoviridae, and Microviridae) that were fully annotated at the family level in both the DZ and DY groups were selected for phylogenetic analysis. The Genomoviridae family currently includes 10 genera and consists entirely of non-enveloped viruses with single-stranded DNA genomes of approximately 1.4–2.4 kb. A total of 375 distinct vOTUs from this family were identified across both groups. Phylogenetic analysis was performed on sequences longer than 3,000 bp (Fig. 8A), showing that these vOTUs exhibited genetic overlap with viruses infecting chordate hosts, including humans (NC_038497) and avian species (NC_076436, NC_033270, NC_035477). Fourteen unique vOTUs were identified from Parvoviridae family. Phylogenetic analysis of sequences exceeding 3,000 bp (Fig. 8B) indicated that these vOTUs clustered primarily with viruses isolated from bats and Culex pipiens. They showed only limited genetic relatedness to those infecting humans, poultry, and livestock. The Smacoviridae family currently encompasses 12 genera and features single-stranded DNA genomes measuring 2.3–2.8 kb. In this study, we identified 283 distinct vOTUs assigned to this family. Phylogenetic analysis based on sequences longer than 3,000 bp (Fig. 8C) demonstrated strong genetic relatedness among these vOTUs, as well as a degree of similarity with viruses known to infect humans and domestic animals. Evolutionarily, this connection implies a possible capacity for interspecies transmission. The Inoviridae family currently includes 26 genera and harbors circular single-stranded DNA genomes ranging from 4.5–8 kb. The Microviridae family consists of two subfamilies and seven genera, with genome sizes between 4.4 and 6.1 kb. Together, these two phage families accounted for 3,179 unique vOTUs. Phylogenetic analysis (Figs. 8D and 8E) demonstrated strong interrelationships among these vOTUs and notable similarities with bacteriophage families known to infect bacteria, highlighting their broad distribution and close evolutionary ties with bacterial hosts.

Figure 7 Prediction of hosts for viral operational taxonomic units (vOTUs).

Sankey diagrams depict the associations between the top five viral families and their potential hosts in the Large-billed crow group (A) and the Northern raven group (B).

Figure 8 The maximum likelihood phylogenetic tree shows the genetic relationships of five virus families: Genomoviridae (A), Parvoviridae (B), Smacoviridae (C), Inoviridae (D), and Microviridae (E).

Red markings denote the vOTUs identified in this study. Different colored modules indicate the host groups infected by these viruses. Numbers beneath branches represent bootstrap support values based on 1,000 replicates.

Discussion

Despite extensive research on the gut viromes of various bird species, including Zhang Wen’s team’s comprehensive analysis of 3,182 birds from eight provinces in China that yielded significant baseline data on avian viral communities (Shan et al., 2022), investigations into scavenger birds—particularly facultative scavengers—remain limited. It is noteworthy that crows carry a variety of parasites, viruses, and bacteria, rendering them crucial as both infection reservoirs and transmission vectors (Fawaz et al., 2016). Consequently, this study utilized viral metagenomics to investigate the fecal DNA viromes of two scavenger species: the Large-billed crows and Northern ravens. Given that the sample size in this study was limited and there was an imbalance in the number of samples between the two crow species, the statistical power may have been low, which could affect the ability to detect true inter-group differences. Therefore, the observed differences should be interpreted as exploratory findings. Initially, in terms of viral family distributions, Genomoviridae constituted the largest proportion of identified DNA viruses in the DZ group. It was not only found in fungal, animal, human, and environmental samples, but also detected in various plant taxa (Nery et al., 2023). A previous study using viromic approaches analyzed viral communities in cloacal samples from wild birds (such as egrets, cranes, and passerines) across multiple nature reserves in China, and identified a large number of viruses belonging to the family Genomoviridae. This suggests a widespread presence of these viruses among wild bird populations. Notably, the authors pointed out that some of these viruses may originate from fungi ingested by the birds or other microorganisms present in their environment (Yao et al., 2022).

By comparison, Parvoviridae emerged as the predominant DNA viruses in the DY group. These small single-stranded DNA viruses form one of the most diverse and widespread viral groups, with 62 known species capable of infecting both vertebrates and invertebrates. Their genomes, roughly five kb in length, mainly encode non-structural proteins (NS) and capsid proteins (VP) (De Souza et al., 2018). Featuring compact genomes and high mutability, they demonstrate notable adaptability to diverse hosts and environments, infecting a wide range of animal species such as cattle, dogs, bats, rodents, and non-human primates. Notably, parvovirus B19 is associated with conditions like fetal hydrops, erythema infectiosum in children, arthritis in adults, and aplastic crises in individuals with hemoglobin disorders (Li et al., 2023). A study investigating the viruses harbored by the snow chough (Chionis albus) revealed that Parvoviridae were predominant within the DNA virus community (Zamora et al., 2023). Though the relative abundance of Parvoviridae was not quantified, the virome data distinctly identified members of this family, suggesting their origin was linked to scavenging on marine mammal (such as seal) carcasses.

Furthermore, a study on the fecal virome of wild migratory ducks revealed the diversity, species-specific differences, and potential impacts on public health and animal health of these birds’ viral communities, suggesting that differences in viral communities could be linked to feeding behaviors and ecological niches (Ramirez-Martinez et al., 2018).

Concurrently, a study examining the gut microbiota composition of Grus monacha proposed that its fish-dominated high-protein diet might indirectly boost the abundance and diversity of intestinal viruses (phages) through the enrichment of certain bacteria (like Cetobacterium and Escherichia) (Takada et al., 2024). By comparing our findings, an initial deduction was made: dietary differences and environmental factors could potentially explain the varying abundance patterns of Genomoviridae and Parvoviridae in the viral communities of the two crow groups. Nevertheless, this conclusion remained a speculative hypothesis based on preliminary observations and existing literature. Although prior studies had not confirmed that diet influences the viral community composition in wild birds, research indicated that changes in the diet of large noctule bats directly influenced the seasonality in the types and abundances of RNA viruses they carried, where higher dietary diversity correlated with increased viral diversity; similar prey compositions resulted in similar viral communities (Huang et al., 2024). Regrettably, limited by the small sample size of this study, it was challenging to attribute the abundance differences in the viral communities of the two crow species directly to feeding amounts, environmental factors, or specific food types. Future work should involve using amplicon sequencing technology for detailed dietary analysis of crows and integrating these data with virome data through correlation or modeling approaches, which would help ascertain whether dietary differences act as critical determinants of the observed viral community composition trends between the two groups.

A study analyzing the viromes of 3,404 wild birds from five provinces in China found that Genomoviridae and Parvoviridae were notably abundant among Passeriformes (Dai et al., 2024). In our investigation, these two viral families constituted a significant portion in both the DZ and DY groups, indicating that Passeriformes birds may act as crucial natural reservoirs for these viruses. Additionally, Caudoviricetes_Unclassified constituted a substantial part of the viromes in both DZ and DY groups, underscoring the importance of the Caudoviricetes taxa in the two crow viromes, even though they had not been fully classified. Indeed, there were also several virus families with small abundance that had not been fully classified in both groups, which limited our deeper understanding of the viral communities carried by the two types of crows. Although a proportion of the viruses remained unclassified, the characterization of viral communities in both crow groups distinctly illustrated the diversity of DNA viruses present in their fecal samples.

Overall, the study’s findings on viral species identification at both the phylum and family levels revealed a high degree of similarity among the DNA viruses found in the feces of the two crow species. This similarity is evident in the highly consistent composition of viral phyla at the phylum level and the presence of numerous core viral families in both groups at the family level, although there are variations in their specific relative abundances. This pronounced compositional similarity corresponds with the shared habitats and significantly overlapping ecological behaviors of the two crow species, especially their omnivorous and scavenging traits (Maeda et al., 2013). Likewise, Lu et al. (2022) reported in their investigation of the intestinal viromes of two endangered lizard species (Phrynocephalus erythrurus and P. theobaldi) on the Qinghai-Tibet Plateau that despite these species being taxonomically distinct, their viral communities exhibited high compositional similarities, dominated by Caudovirales. They suggested that this convergent viral profile likely stemmed from the shared extreme environmental conditions and dietary habits of the two lizard species. Notably, this observed similarity contrasts somewhat with our initial expectations of viral community differentiation based on sampling sites (spanning various altitudinal gradients), suggesting that in shaping the overall structure of fecal viral communities, host ecological habits and environmental factors might exert a stronger influence than the altitude gradients assessed in this study. It should also be acknowledged that the limited sample size of this study restricts robust statistical validation, indicating that the observed patterns of high similarity warrant further exploration in larger-scale studies. Nonetheless, this pioneering systematic characterization of the fecal virome compositions in two crow species retains important preliminary significance.

Phages play a pivotal role in modulating bacterial diversity and functionality (Bonilla-Rosso et al., 2020); being viruses that infect bacteria and archaea, they constitute one of the most plentiful elements within the microbiome, particularly amongst DNA viruses. In our study, the characterized viral communities were predominantly comprised of phages. The identified phages primarily originated from Uroviricota, necessitating future experimental designs to elucidate the cause of this predominance. In our investigation, predictions about the bacterial hosts of phage contigs were made; to further investigate the interplay between phages and their hosts, it is necessary to design experiments to interpret these interactions.

Through comparisons across multiple databases, functional insights into the viruses harbored by the two crow species were gained. The KEGG annotations revealed an enrichment primarily in metabolic pathways and genetic information processing for both crow groups, aligning with findings from the fecal virome of red-billed gulls, where similar enrichments in these pathways were observed (Liao et al., 2023). Interestingly, virus functional annotations in reptiles also closely mirror our findings. Li et al. (2025) reported in their analysis of the gut virome of the sea turtle Chelonia mydas that viral genes were notably enriched in metabolic pathways and genetic information processing categories, suggesting that these functionalities may reflect interactions between viruses and host microbiota, contributing to the maintenance of intestinal microecological stability in Chelonia mydas. In our investigation, replication and repair processes within genetic information handling were prominently enriched in both groups, possibly linked to the increased risk of pathogen exposure while consuming carcasses, supporting genetic stability and adaptability within the microbiome. This observation corroborates our prior findings on functional pathways of DNA viruses in the gut of Himalayan vultures (Zhai et al., 2023). Assuming these scavenging birds ingest pathogen-laden carrion, elevated expression of replication and repair genes would aid in preserving genomic stability, mitigating the risk of DNA damage due to environmental pressures and pathogenic attacks. DNA replication is essential for genome integrity (Saxena & Zou, 2022).

Furthermore, a study focusing on antibiotic resistance genes (ARGs) in the feces of various wild bird species confirmed that ARG abundance is significantly correlated with the geographical and ecological conditions of bird habitats (Luo et al., 2022). Concurrently, a study on the gut microbiota and ARGs of Anser erythropus revealed that differences in food sources and environmental pollution levels across different wintering areas led to significant variations in ARG abundance and types within this species (Liu, Xu & Feng, 2023). Recent research on antibiotics highlighted that sources of antibiotics in the environment include hospitals, livestock farming, wastewater treatment facilities, and other pollution sources (Lu et al., 2025). Cao et al. (2020) demonstrated in their study on migratory birds that those with complex dietary sources carry a greater diversity of ARG types. In our study, it was found that the DY group exhibited a wider variety and even distribution of ARG types (12 types, with the highest proportion at just 18.53%), whereas the DZ group was predominantly characterized by polymyxin resistance genes (69.92%). This indicates that birds in the DY group may have encountered more varied and complex environmental stresses, potentially linked to richer food resources and diverse pollution sources in their habitat. Conversely, the dominance of polymyxin resistance genes in the DZ group suggests specific selective pressures, such as particular antibiotic use or pollution sources in that region. Our findings support previous conclusions that the abundance and types of ARGs in wild birds are closely associated with their geographical and ecological conditions, food sources, environmental pollution levels, and specific pollution sources. However, it is important to note that while the aforementioned analyses provide valuable insights into host ecology, the conclusions drawn from functional analyses and ecological inferences remain preliminary hypotheses that require further validation through additional studies. Specifically, the potential of these two crow species as carriers of ARGs warrants further investigation and confirmation.

It is necessary to recognize several limitations of this study. Firstly, the small sample size poses a major constraint. Collecting fecal samples from Northern ravens and Large-billed crows in the wild, particularly fresh samples, presents considerable logistical challenges. This limited sample size markedly reduced the statistical power of our analysis, thereby undermining the confidence in the observed differences in viral abundance between groups. Such low statistical power significantly compromises the generalizability of our findings. As a result, all comparisons of viral abundance and between-group differences reported here should be interpreted as exploratory and preliminary in nature. Future studies should aim to include larger sample sizes and wider geographic representation to enhance the robustness of conclusions.

Additionally, the viromic approach itself has inherent limitations. One major issue is its relatively low sensitivity in samples with low viral loads (Gauthier et al., 2023). In our case, this may result in underrepresentation of low-abundance viral taxa or phages associated with rare bacterial hosts, potentially leading to an underestimation of overall viral diversity. Furthermore, inaccuracies in estimating viral relative abundance could affect the interpretation of abundance differences between groups—such as variations in the proportions of specific viral families. Virological research has been heavily focused on human and mammalian viruses (Santiago-Rodriguez & Hollister, 2023), which makes the analysis of viromes from atypical hosts, such as birds, particularly challenging. This is because closely related reference sequences are often scarce or absent in public databases. Consequently, we could only identify those viral sequences already present in existing databases, leaving many others unclassified. Moreover, the accuracy of viral identification is highly dependent on the comprehensiveness and quality of current reference databases (Rose et al., 2016). In our study, a number of viral sequences could not be confidently assigned to the family level or finer taxonomic classifications, which substantially limited our ability to fully resolve the composition of the intestinal viromes in the two crow species.

Additionally, the workflow complexity introduced further challenges. Despite our best efforts to minimize contamination during sample collection, it is difficult to completely rule out the possibility of sample contamination. Additionally, variations in bioinformatics tools and analytical methods can introduce biases during data processing. Viral metagenomic analysis involves a series of complex bioinformatic steps, including assembly, contamination removal, and annotation. The choice of software, parameter configurations, and filtering criteria can all influence the final results (Santiago-Rodriguez & Hollister, 2023). Although we followed established protocols, systematic biases introduced during these steps cannot be completely eliminated. Furthermore, this study employed the REPLI-g Single Cell Kit (QIAGEN), which is based on the principle of multiple displacement amplification (MDA). This method has been shown to exhibit a significant amplification bias toward single-stranded DNA (ssDNA) and circular genomic templates (Estevez-Gomez et al., 2025; Lasken & Stockwell, 2007). This suggests that the relative abundance of ssDNA viruses observed in our study may be systematically overestimated, while the true proportion of dsDNA viruses may be underestimated. However, as our study primarily focuses on the structural composition and relative quantification of the viral community, this amplification bias has a minimal impact on the core conclusions of the study, although it may affect the accuracy of absolute quantification results. Among the listed limitations, the small sample size notably affected the statistical robustness of our findings. Nevertheless, this study still offers valuable preliminary data regarding viral community structure, potential virus-host interactions, and ecological functional inferences. Follow-up studies with larger sample sizes will help mitigate some of the current limitations and may reveal novel insights. Overall, despite the unavoidable constraints on sample size, our work still contributes to the understanding of the fecal viromes of these facultative scavenging avian species.

Conclusions

In conclusion, this study employed viral metagenomics to offer preliminary insights into the fecal DNA viromes of the Large-billed crows and Northern ravens characterized and conducted detailed quantifications of the types and abundances of DNA viruses they harbor, and additionally analyzed the phage diversity within them. The findings highlighted the differences and similarities in viral community composition and functionality between the two crow species, thus laying the groundwork for future studies and offering critical data support. Future investigations will be able to further clarify the intricate interplay among viruses, hosts, and their environments. In view of the growing attention to the potential role of wildlife in the transmission of zoonotic diseases, our findings highlight the importance of expanding surveillance efforts and enhancing monitoring of viral diversity in bird populations, particularly those that have close contact with human settlements. Establishing an early warning system and strengthening cross-species pathogen tracking mechanisms will help improve preparedness and response capabilities against potential zoonotic disease outbreaks.

Supplemental Information

Supplemental Information 1 Raw data after sequencing.

Used to describe the quality of sequencing data, including sample ID, total number of reads, total base count, number of reads containing “N” and their percentage, as well as the percentage distribution of A, T, C, G, and N bases, error rate, Q20%, Q30%, and GC content.

Supplemental Information 2 Clean data after quality control.

The data after quality control, including total number of reads, total base count, number of reads containing “N” and their percentage, base composition (A%, T%, C%, G%, N%), sequencing error rate, proportion of high-quality bases (Q20%, Q30%), and GC content.

Supplemental Information 3 Clean data after rmove host.

The statistical information of sequencing data after host screening, including sample ID (SampleID), total input paired reads (TotalInputReadsPair), host-derived paired reads (host), non-host derived paired reads (nohostReadsPair), and the proportion of non-host paired reads relative to total input paired reads (nohostrate%).

Supplemental Information 4 Data after assembling the reads with host sequences removed

Statistical information of the assembled sequences, including sample ID (SampleID), number of sequences (Number), total sequence length (Length), N50 and N90 metrics (N50, N90), maximum sequence length (Max), and minimum sequence length (Min).

Supplemental Information 5 Number of viral candidate sequences identified using IMG, deepVirFinder, VirSorter, and Vibrant

The number of candidate viruses identified using different software.

Supplemental Information 6 Data on the detection results of antibiotic resistance genes in the Large-billed Crow group and the Common Raven group

The detection results of different types of antibiotic resistance genes.

Supplemental Information 7 Venn diagram showing the number of viral candidate sequences identified by IMG, deepVirFinder, VirSorter, and Vibrant

The number of virus candidate sequences identified by different software: IMG identified 1,200 candidate sequences, Virome Sorter identified 29,795 candidate sequences, VIBRANT identified 20,446 candidate sequences, DeepFinder identified 27,127 candidate sequences.

Supplemental Information 8 A pie chart shows the quality and completeness of vOTU sequences evaluated by CheckV

The sequence quality assessed by CheckV, where different colors represent the percentage of sequences at different quality levels, including high quality (3.51%), medium quality (7.17%), low quality (61.63%), and so on.

Additional Information and Declarations

Competing Interests

Author Contributions

Animal Ethics

Data Availability

The authors declare there are no competing interests.

Yonggang Dong conceived and designed the experiments, performed the experiments, analyzed the data, prepared figures and/or tables, authored or reviewed drafts of the article, and approved the final draft.

Sitong Fan conceived and designed the experiments, performed the experiments, analyzed the data, prepared figures and/or tables, and approved the final draft.

Shunfu He conceived and designed the experiments, performed the experiments, analyzed the data, prepared figures and/or tables, and approved the final draft.

Wenxin Zhao conceived and designed the experiments, performed the experiments, analyzed the data, prepared figures and/or tables, and approved the final draft.

Zhuoma Lancuo conceived and designed the experiments, performed the experiments, analyzed the data, prepared figures and/or tables, and approved the final draft.

Kirill Sharshov conceived and designed the experiments, performed the experiments, analyzed the data, prepared figures and/or tables, and approved the final draft.

Ying Li conceived and designed the experiments, performed the experiments, analyzed the data, prepared figures and/or tables, authored or reviewed drafts of the article, and approved the final draft.

Wen Wang conceived and designed the experiments, performed the experiments, analyzed the data, prepared figures and/or tables, authored or reviewed drafts of the article, and approved the final draft.

The following information was supplied relating to ethical approvals (i.e., approving body and any reference numbers):

This study conformed to the guidelines for the care and use of experimental animals established by the Ministry of Science and Technology of the People’s Republic of China (Approval number: 2006-398). The research protocol was reviewed and approved by the Ethical Committee of Qinghai University. This study did not involve capture or any direct manipulation or disturbance of Large-billed crows and Northern ravens.

The following information was supplied regarding data availability:

The raw sequence data are available in the Genome Sequence Archive in National Genomics Data Center, China National Center for Bioinformation/Beijing Institute of Genomics, Chinese Academy of Sciences: CRA025033.

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
