# Peer review of "Comparative analysis of fecal DNA viromes in Large-billed crows and Northern ravens reveals diverse viral profiles"

_PeerJ, doi:10.7717/peerj.20170_

## Round 0.1 · original submission · Major Revisions

Both reviewers provided helpful feedback for improving the manuscript. I do not feel that any additional samples need to be analyzed, but do suggest you make all changes in the text that they recommend. In addition, since these are fecal samples, it would be preferable to use the wording "fecal viromes" instead of "gut viromes". Finally, I note that REPLI-g was used; does that method bias towards ssDNA/circular templates? That may affect the ability of this method to yield quantitative results.

Reviewer 1 ·

Basic reporting

The quality of the figures is to be revised.

Experimental design

-

Validity of the findings

-

Additional comments

The manuscript “Comparative analysis of gut DNA viromes in large-billed crows and northern ravens reveals diverse viral profiles” explores an important topic and uses relevant tools to analyze gut viral diversity in two crow species. This study uses metagenomics to analyze DNA viruses present in the gut of two facultative scavengers and aims to assess viral diversity and the composition of these viral communities to evaluate the zoonotic risk through these birds, which were chosen as subjects due to their wide distribution, proximity to humans, and unique behaviour.

One of the biggest values of this MS is that it offers a fresh look at a very relevant subject in bird ecology and the use of modern methodologies for the analysis.

Although it is not a novel topic of investigation, this study still provides new data on a well-known corvid species, which is very interesting to ecologists and health professionals. This MS identifies different viral families, such as Genomoviridae and Parvoviridae, and the findings show that viral diversity is similar between the two different bird species, as well as similar viral composition between these species. The authors offer interesting insights into the gut viromes of both the Large-billed crow and the northern raven, opening the door to future investigations.

Based on the reviewer's opinion, this paper is recommended for publication after some revisions are made.

General comments

In the MS, with a brief conclusion, we get some key points on the topic as the authors resumed the study in a very comprehensive manner. The conclusion reflects well on the objectives, which are supported by the presented results, and includes potential future directions. Yet, the conclusion could be enriched by discussing the prospects for zoonotic surveillance, or the measures to be taken in the face of the risks detected.

The title is directly referenced in the manuscript; brief, and it clearly defines the topic discussed.
The abstract is brief and summarises the essence of the MS. However, there are some parts that the reviewer wishes the authors to revise. For instance, the methodology is hardly mentioned; it would be beneficial to describe some details like samples, the number of birds studied, and the type of sequencing used. Additionally, although the results are presented in the most significant and striking manner, statistical reporting would add value to the abstract.

The introduction is sufficiently concrete. Providing readers with comprehensive and up-to-date information on all aspects of the study, with well-described objectives, although, authors’ predictions are not defined. Still, the subject of the study was widely introduced and well-explained. The authors cited 50 papers, which were mostly published recently. This illustrates how the papers were carefully chosen and how pertinent the research topic is.

The Methods and Materials section is well established. The techniques, protocols, and programs used were described in enough detail for the reader to follow each part of the MS and easily understand the findings. However, a small paragraph describing the species in question and the area of study would be a beneficial addition, as it is probable that not all readers may have a background in corvid ecology. Also, providing data on the local populations can be useful.

The results section is very insightful, with several figures showcasing the findings. Additionally, the findings are displayed in a logical sequence, making it easy for the readers to relate the results to the previous sections of the paper. However, in most of this section, the textual explanation describes all the findings, which is a kind of redundancy. This can be overwhelming to the readers. Since the authors chose to use figures to display their findings, the textual explanation should describe specific observations and unique findings that are particularly abundant, novel, or have predicted functional significance only. In the result section, we choose Clarity over Quantity, as it is important not to overload the reader. Therefore, Information presented in tables or figures should not be repeated in the text unless necessary for emphasis or explanation.

Additionally, the results chapter should present only the data and findings. Authors must avoid any interpretation or discussion of their findings, as these belong in the discussion section. For instance, Lines 256-259, Lines 263-264, Lines 294-300, Lines 376-378, and Lines 381-386 are interpretations of results, which belong to the discussion chapter.

Authors should also refrain from repeating details about the methods used in the results chapter, as is the case in Lines 217-218. It is better to refer back to the methods section if necessary.

The discussion part is very informative, the authors made sure to highlight their previous knowledge as well as the richness of their literature review in regard to their study premises. However, a discussion should be more concise, and here, a big part reflects more on the introduction of the topic rather than the interpretation and discussion of the results, resulting in some unnecessary repetitions in the manuscript, for instance, in lines 400-412 the reader is presented with a topic description rather than result discussion, this part should be in the introduction. Additionally, in lines 422-423, the authors referred to their hypothesis for the first time in the manuscript. Authors should express their predictions in the introduction chapter first.

Specific Comments
Line 56: a dot is missing at the end of the sentence.
Line 80: Please add a reference to the statement.
Line 93: “E. coli” should be in italic as it is a species name.
Line 112: a dot is missing at the end of the sentence.
Line 450: The reference cited does not seem to reflect on the statement; please revise.

Despite the above comments, which are intended as constructive criticism, the reviewer, by no means, undervalues the quality of the work performed by the authors. These findings are believed to be valuable.

Questions:
1. How did the observers assess the physical conditions of the birds? What were the criteria used, since the birds were not captured?

2. What is the origin of the protocol used for the concentration and enrichment of virus particles? Was it optimized by the authors, or was it obtained from a previous study? If so a reference should be cited.

Reviewer 2 ·

Basic reporting

This manuscript presents an interesting and timely study on the gut DNA viromes of two crow bird species, providing valuable baseline data on their viral communities through viral metagenomics. The topic is relevant given the ecological importance of facultative scavenger birds as potential reservoirs and vectors of viruses. The authors offer thoughtful discussion linking viral diversity to feeding behavior and functional pathways, contributing to our understanding of avian viromes.

However, the Introduction lacks a clear articulation of the study’s hypotheses. While multiple hypotheses appear later in the manuscript, they are not clearly introduced at the outset. Moreover, there is no mention of bacteriophages or their potential role, despite the fact that the Discussion devotes a substantial section to phage research. The Introduction should at least allude to the significance of phages in bird gut viromes to provide coherence and better prepare the reader for the discussion that follows.

Similarly, the manuscript introduces antibiotic resistance genes (ARGs) and their potential spread via scavenger birds only in the Discussion, with no prior context in the Introduction. This thematic disconnect can be confusing and makes the Discussion feel like it is introducing entirely new concepts rather than building upon the study’s stated objectives.

Additionally, the manuscript occasionally suffers from complex sentence structures that impair readability. There also appears to be a taxonomic error in the identification of one species. The manuscript refers to the Northern Raven as Corvus cayanoides, which is not a recognized scientific name. The correct name is Corvus corax, and the manuscript should be updated accordingly for taxonomic accuracy.

On the other hand, in the section Discussion limited comparison with other studies/species, and more context would strengthen impact. Additionally, while the manuscript notes some methodological constraints of viral metagenomics, it would benefit from a clearer discussion of how these may impact the data interpretation. The functional analyses and links to host ecology remain largely speculative and should be framed accordingly.

Experimental design

The methodology employed by viral metagenomics, combined with current bioinformatics pipelines, is appropriate and well-aligned with the study’s objectives. The authors clearly explain the analytical methods, including host prediction and phylogenetic analysis, and identify key findings related to viral diversity and host relationships. The potential for cross-species transmission is a relevant and timely consideration in virology, and its inclusion is commendable.

However, important methodological details are missing. There is no information on the time of year when samples were collected nor on the sampling protocol itself. It is unclear how the researchers confirmed that fecal samples came from the intended bird species, or how they avoided contamination. The supplementary image shows that sampling sites are geographically dispersed, with one at a high-altitude location and another at a lower elevation. Nonetheless, the manuscript does not elaborate on these sampling sites, which is critical given the consistent reference to dietary differences between the two bird species as a factor influencing gut virome composition.

Validity of the findings

The study provides valuable preliminary insights into viral family composition differences. The results regarding differences in viral family composition are valuable, and the application of viral metagenomics is well-justified, particularly for exploring potential zoonotic risks.

However, I would like to raise an important concern about the total sample size used in the study. It is very limited, as the authors analyzed only seven individuals from two species (four large-billed crows and three northern ravens). While the sequencing and analysis are well presented, the small sample size severely limits the statistical power of the analysis. A very limited sample size is too small for robust statistical analysis, meaning it is not possible to draw any meaningful or significant conclusions or detect any significant differences. This restricts the strength of the conclusions. Since the statistical test will have very low power, statistical comparisons and claims of differential abundance should be treated as exploratory and interpreted with caution. The authors should avoid overgeneralizing their findings.

Additional comments

The manuscript would benefit from a more extensive comparison with related studies and a stronger contextualization of findings within the broader literature. While the authors acknowledge some limitations of viral metagenomics, a more detailed discussion on how these constraints may affect data interpretation would improve transparency.

Additionally, the functional analyses and ecological inferences drawn from the viromes data remain speculative and should be framed accordingly. The discussion around links between virome composition and host ecology, while intriguing, should be presented as preliminary.

Additionally, the Discussion section has very limited comparison with other similar and current studies or species, and more context would have strengthened the impact by highlighting how the findings align with or differ from existing knowledge, and by providing a broader scientific perspective.
Overall, the manuscript addresses the issues outlined above particularly those related to methodological transparency, taxonomic accuracy, and cautious interpretation and thus this work has the potential to make a meaningful contribution to the field.

I provided my additional comments in the PDF document.

Annotated reviews are not available for download in order to protect the identity of reviewers who chose to remain anonymous.

---

## Round 0.2 · Minor Revisions

Thank you for revising the manuscript to address the reviewers' concerns. Upon my own thorough review of the revision, I have a few minor suggestions that should be incorporated before the manuscript can be accepted.

1) In the title, remove "the"
2) In the title and throughout the paper, be consistent with capitalization of ravens and crows
3) Anywhere you refer to formal virus taxonomy (e.g., a family level), this should be italicized (https://ictv.global/faq/names)
4) Remove the sentence about NCLDVs (lines 54-56), it is out of place
5) In general, the paragraphs are too long and the text should be divided so each paragraph addresses only a single topic. As an example, I would create a new paragraph on line 78 (where you switch from talking about zoonotic disease to talking about dietary preferences), and on line 106 when you start talking about phage and ARGs. Same in the discussion (e.g., line 449 where you start talking about Parvoviridae, and line 469 where you talk about dietary differences, and line 543 where you start talking about ARGs)). This will help break the text into more manageable chunks and be easier for readers.
6) On line 125, change "Metaviomes" to "metaviromics"
7) On line 199, define EP tubes
8) In the section starting on line 392, why are the antibiotic names italicized?
9) When introducing non-bird species (e.g., Chelonia mydas on line 532), state what type of animal (e.g., the sea turtle Chelonia mydas)
10) There are several inconsistencies in the reference section (e.g., title capitalization, typing errors, look through carefully.
11) Similar to the reviewers of the previous version, I find the revised figures in the Reviewer PDF to be impossible to read. I don't know if this is due to a quality issue in the figures themselves, or was caused by compiling the PDF.

---

## Round 0.3 · accepted · Accept

Thank you for your careful attention to making the previously noted revisions to the manuscript, and congratulations!